# Identification of Divergent Isolates of Banana Mild Mosaic Virus and Development of a New Diagnostic Primer to Improve Detection

**DOI:** 10.3390/pathogens9121045

**Published:** 2020-12-12

**Authors:** Marwa Hanafi, Rachid Tahzima, Sofiene Ben Kaab, Lucie Tamisier, Nicolas Roux, Sébastien Massart

**Affiliations:** 1Integrated and Urban Plant Pathology Laboratory, Gembloux Agro-Bio Tech, University of Liège, 2, Passage des Déportés, 5030 Gembloux, Belgium; rachid.tahzima@doct.uliege.be (R.T.); sofiene.benkaab@doct.uliege.be (S.B.K.); lucie.tamisier@uliege.be (L.T.); sebastien.massart@uliege.be (S.M.); 2Consultative Group on International Agricultural Research, 34090 Montpellier, France; n.roux@cgiar.org

**Keywords:** high throughput sequencing (HTS), diagnostic, BanMMV, Betaflexiviridae, primers, novel isolates, plant virus, total RNA

## Abstract

*Banana mild mosaic virus* (BanMMV) (Betaflexiviridae, Quinvirinae, unassigned species) is a filamentous virus belonging to the Betaflexiviridae family. It infects *Musa* spp. with a very wide geographic distribution. The genome variability of plant viruses, including the members of the Betaflexiviridae family, makes their molecular detection by specific primers particularly challenging. During routine indexing of the *Musa* germplasm accessions, a discrepancy was observed between electron microscopy and immunocapture (IC) reverse transcription (RT) polymerase chain reaction (PCR) test results for one asymptomatic accession. Filamentous viral particles were observed while molecular tests failed to amplify any fragment. The accession underwent high-throughput sequencing and two complete genomes of BanMMV with 75.3% of identity were assembled. Based on these sequences and on the 54 coat protein sequences available from GenBank, a new forward primer, named BanMMV CP9, compatible with Poty1, an oligodT reverse primer already used in diagnostics, was designed. A retrospective analysis of 110 different germplasm accessions from diverse origins was conducted, comparing BanMMCP2 and BanMMV CP9 primers. Of these 110 accessions, 16 tested positive with both BanMMCP2 and BanMMV CP9, 3 were positive with only BanMMCP2 and 2 tested positive with only BanMMV CP9. Otherwise, 89 were negative with the two primers and free of flexuous virions. Sanger sequencing was performed from purified PCR products in order to confirm the amplification of the BanMMV sequence for the five accessions with contrasting results. It is highly recommended to use the two primers successively to improve the inclusiveness of the protocol.

## 1. Introduction

Banana (*Musa* spp.) is a perennial herbaceous plant and is among the most important staple food crops worldwide. It is also considered to be a substantial source of income for producers [1]. With worldwide annual production reaching 144 million tons, it represents an important contribution to the economies of many countries in Asia, Africa, Latin-America and the Pacific Islands [2]. Bananas are vegetatively propagated, which involves a high risk of virus spread through the movement of infected planting material [3].

There is a requirement for virus-tested planting material for guaranteeing the exchange of germplasm and for disease control [4]. In this context, the International Plant Protection Convention (IPPC), a plant health treaty signed by over 180 countries, was developed in the 1920s in order to address organisms that are both directly and indirectly harmful to plants [5]. The reliable detection of plant pathogens is therefore a crucial step in the proper management of many diseases, and to avoid their geographical extension due to exchanges of planting material.

However, some problematic situations make the detection of viruses in banana more difficult and therefore time-consuming. For instance, viruses that may be symptomless or at a very low titer in in vitro plants or young plantlets pose a special threat to the germplasm distribution [6]. Alternatively, the presence of endogenous infective *Banana streak viruses* (eBSV) in almost all B genome-containing *Musa* cultivars makes the detection process trickier, since it requires the distinction between episomal and endogenous viral sequences [7]. Furthermore, viruses that have high genome variability, such as *Banana mild mosaic virus* (BanMMV), make molecular detection particularly challenging. The latter virus displays an important degree of genetic diversity [3,8]. For example, divergent isolates of grapevine virus T, another member of the Betaflexiviridae family, presented nucleotide mismatches at the primer-binding site, which could hamper their detection [9,10]. Therefore, some BanMMV isolates could be missed by very specific diagnostic primers, resulting in false-negative results.

BanMMV is an unassigned member in the Betaflexiviridae family [11]. Its complete genome consists of 7352 nucleotides, encoding five open reading frames (ORFs) [4]. BanMMV is present in most, if not all, banana-producing areas of the world, in South America, Asia, Africa, Oceania and Australia [8,12]. Usually, infected plants are symptomless. However, transitory symptoms, such as chlorotic streaks and mosaic, are occasionally observed on young leaves, but often disappear as the plant matures [8]. Gambley and Thomas (2001) and Sharman et al. (2001) have reported that some more pronounced symptoms of BanMMV infection were associated with a limited number of cultivars, including cv. ‘Ducasse’ (syn. ‘Pisang Awak’, ABB genome). BanMMV often occurs as a mixed infection with other banana-infecting viruses such as banana streak viruses (BSV) and cucumber mosaic virus (CMV), and the mosaic symptoms of BanMMV may be masked by those of the other viruses [1,13,14]. On the other hand, previous studies reported that, in Guadeloupe, infection by BanMMV exacerbated the symptoms of CMV causing leaf necrosis symptoms compared with mosaic symptoms from CMV infection alone [8].

BanMMV is only known to infect species from the *Musa* genus. BanMMV is mainly transmitted through tissue culture or vegetative propagation. No natural vector has been identified and mechanical transmission has not been successful. Virus transmission attempts made from a single infection in cv. Ducasse failed with mealybugs and aphids, via soil collected from around infected plants, and by root-to-root contact [8].

Several diagnostic protocols have been published for BanMMV. Monoclonal and polyclonal antibodies developed against BanMMV [8,13] can be used for the detection of the virus by ELISA. Monoclonal antisera were unsuitable for the routine indexing of BanMMV since they were greatly specific to the viral strain that was used to produce them, while BanMMV has shown a very high level of genome variability [11,13]. However, polyclonal antisera were suitable and successfully used for immunosorbent electron microscopy (ISEM) and in immunocapture (IC) reverse transcription (RT) polymerase chain reaction (PCR) [13]. A nested IC-RT-PCR assay has been described [13] for BanMMV detection, targeting the RNA-dependent RNA polymerase (RdRp) region. It is based on polyvalent degenerate primers (PDO) developed by the authors of Ref. [15]. Downstream (toward the 3′ end) primers targeting the coat protein (CP) gene have been also developed to recognize BanMMV isolates [3,8].

One approach to minimize this issue is the inclusion of degenerated bases within the oligonucleotide primers. However, a primer with a high level of degeneracy might lead to a loss of primer template-specificity [16,17].

High-throughput sequencing (HTS), also called next-generation sequencing (NGS) or deep sequencing, has proposed a solution to these issues and thus revolutionized plant virus diagnostics. The availability of these powerful tools offers new opportunities and possibilities in routine diagnostics as they enable the simultaneous sequencing and detection of multiple viruses in a sample, regardless of their genome sequence and, therefore, without any *a priori* knowledge [18]. Currently, they are accelerating the identification of viruses associated or not with a disease of unknown etiology [19,20,21]. This allows unbiased and hypothesis-free testing in particular for symptomless plant material [22].

This study aimed to describe the identification and genome sequencing of two isolates of BanMMV, and, based on the virus sequences available in GenBank, to design and test a new diagnostic primer for a routine indexing use. A retrospective analysis of 110 accessions from the international banana germplasm collection was conducted using this newly designed primer to compare its performance with that of an existing diagnostic primer.

## 2. Materials and Methods

### 2.1. Plant Material

Banana accession ITC0763 from the International *Musa* Germplasm Transit Centre (ITC) (Bioversity International) presented discrepancies between diagnostic techniques used for indexing when assessed under post entry quarantine in Australia [23]. No viral symptoms were observed for the accession. To investigate this, three deflasked in vitro plantlets were grown in 25 cm diameter pots in Gembloux, Belgium, initially in a humid chamber (90% relative humidity) at a temperature of 23 ± 2 °C and a 16 h photoperiod, and then after three weeks, in a heated insect-proof greenhouse with the same conditions of temperature and photoperiod.

After 6 months of acclimatization, 1/3 of the youngest leaves were sampled from each plant using a disposable sterile scalpel blade. In the laboratory, 8 discs/leaf samples were subsampled from both laminar and midrib using a water-cleaned then bleached 4 mm leaf punch. These 24 samples were directly processed (but they may also be stored at −80 °C).

One hundred and ten additional banana accessions from different countries and continents were tested in order to validate the new diagnostic primer from this study. They were all grown and sampled as ITC0763. Details are listed in Appendix A. Healthy accessions were also treated as these accessions.

### 2.2. RNA Extraction

Total RNA was extracted from leaf subsamples (100 mg in total) using RNeasy Plus Mini Kit (QIAGEN, Hilden, Germany) according to the manufacturer’s instructions. Purified RNA concentration was quantified by spectrophotometry, and quality was evaluated using the Agilent 2100 Bioanalyzer, Belgium (2100 expert software, ver B.02.07.SI532).

### 2.3. Crude Extracts Preparation

Crude extracts were prepared by grinding the leaf subsamples using a TissueLyser (QIAGEN, Hilden, Germany) in 1 mL of a CMV milk extraction buffer called common extraction buffer (CEB) (0.05 M citrate, pH 8.0, containing 0.5 mM EDTA, 2% PVP, 0.05% Tween-20 and 0.5% monothiogylcerol) [24] and then clarifying the sap extract by centrifuging at 10,000× *g* for 5 min. The supernatant was aliquoted into 500 µL thin-walled polypropylene tubes and stored at −80 °C.

### 2.4. Routine Diagnostic Assays

Routine germplasm screening molecular diagnostic tests were carried out on the crude extracts following the procedure described by [23]. Then, an IC-RT assay was conducted to produce the complementary DNAs, on which the PCR test was carried out. All the details about the molecular diagnostic test used in this study are shown in Appendix A.

### 2.5. Immunosorbent Electron Microscopy (ISEM) of Viral Minipreps

Partially purified and concentrated viral minipreps were prepared as described previously [23]. ‘Necoloidine’ solution (Stanvis)-coated copper grids were coated with a mixture of BSV, BanMMV and BBrMV antibodies before being floated over a droplet of viral miniprep. The prepared grids were stained with 1% ammonium molydbate and viewed using a JEM-1400 TEM (JEOL Ltd., Tokyo, Japan). The micrographs were taken with an ORIUS SC1000 CCD camera (Gatan Inc., Pleasanton, CA, USA). The detailed protocol is described in Appendix A.

### 2.6. Library Preparation and High-Throughput Sequencing

The sequencing library was prepared using the Ribo-Zero™ Plant Leaf Kit (Illumina Inc., San Diego, CA, USA) for ribodepletion (ribosomal RNA depletion) followed by the TruSeq Stranded Total RNA Library Prep Kit (Illumina Inc., San Diego, CA, USA) using the standard protocol as previously described for another virus of Beteflexiviridae [25]. The sample was sequenced on the Illumina Nextseq 500 platform with paired sequencing reads of 2 × 151 nt at the GIGA facilities of Liège University (Liège, Belgium).

### 2.7. Bioinformatics Analysis

The Geneious software v10.2.6 (Biomatters Ltd., Auckland, New Zealand) [26] was used for sequence analysis. The RNA-seq reads were paired, merged using BBMerge from the BBtools suite and the duplicates eliminated. They were then de novo assembled into contigs for genome reconstruction, using the SPAdes software embedded in Geneious with default parameters and a k-mer of 55 [27]. De novo contigs were annotated using the tBLASTx module with the refseq database of viral nucleotides sequences downloaded from GenBank on 15 November, 2019. All the non-redundant reads (merged and unmerged) from the sample were mapped to selected contigs using the Geneious mapper with the following parameters: minimum mapping quality 30, word length 17, maximum mismatches per read 2%, do not allow gaps, minimum overlap identity 98%, index word length 9, maximum ambiguity 16, search more thoroughly for poor matching reads set to yes [28]. Whole genome alignments between our contigs and the refseq sequence of other Betaflexiviridae species were done using multiple sequence comparison by log-expectation (MUSCLE) alignment embedded in Geneious with a maximum number of 8 iterations. SNP calling was performed using Geneious software v10.2.6 with default parameters (Minimum frequency (%) = 0.25, Minimum coverage = 1) in order to determine in which positions there are polymorphisms. To remove possible sequencing artifacts, only the SNPs that appeared at a frequency higher than 1% were retained for further analyses [29].

### 2.8. Phylogenetic and Sequence Analysis

The RdRp and CP genomic regions of the sequenced genomes in this study were retrieved using the ORF finder online tool from GenBank (https://www.ncbi.nlm.nih.gov/orffinder/). In addition to these sequences, the complete genomes of the BanMMV and RdRp nucleotide sequences were aligned and used for phylogenetic analyses with the MEGA software package version 7.0. In addition to these sequences, full nucleotide genomes of BanMMV and other Betaflexiviridae members, RdRp aa sequences of BanMMV and CP nucleotide sequences of the virus were separately aligned using MUSCLE implemented in MEGA, then used for phylogenetic analyses with the MEGA software package version 7.0. [30]. The phylogenetic relationships were inferred using UPGMA embedded in the same version of the MEGA software. The stability of the topology was evaluated using bootstrap (1000 replications) [30,31]. Accession numbers and origins (when available) of virus sequences obtained from GenBank (http://www.ncbi.nlm.nih.gov/) have been integrated in the phylogenetic trees.

### 2.9. Primer Design

Two hundred and ten nucleotide sequences of BanMMV were publicly available in the Genbank database on 16 March 2020. Among them, 54 sequences correspond to the CP sequences of the virus (40 partial and 14 complete sequences), and 154 correspond to partial RdRp sequences. A single complete genome of BanMMV was available before the identification of the two novel genomes from this study.

The new forward primer (BanMMV CP9) was designed via the Geneious v10.2.6. software based on the Muscle Alignment of the 54 CP region sequences, the two novel isolates from this study and the single complete BanMMV genome (GenBank accession NC_002729).

The candidate primer was evaluated for its Tm and self- or heterodimer formation using the IDT OligoAnalyzer 3.1 software (https://eu.idtdna.com/calc/analyzer). A list of all the published primers that could detect BanMMV sequences is represented in Appendix A. The inclusivity of all the published primers, and of the new one, was assessed in silico to test if they are able or not to detect the novel genomes. The analysis for matches was carried out using Geneious 10.2.6 (Biomatters Ltd., Auckland, New Zealand) [32].

### 2.10. Gradient PCR

In order to optimize the annealing temperature of the newly designed forward primer, eight different temperatures ranging from 47 °C to 59 °C were tested, during the gradient PCR process, on healthy and infected banana accessions.

## 3. Results and Discussion

### 3.1. Immunosorbent Electron Microscopy (ISEM) Analysis

Like PCR tests and other methods in plant virus diagnostics, electron microscopy is a specific approach that is based on the use of antibodies as well as knowledge of the target. It can help in the detection of novel viruses. Flexuous rod-shaped virions were detected in a partial purification of banana accession ITC0763 (Figure 1), while negative results were obtained by targeted IC-RT-PCR used to detect filamentous viruses of banana (BanMMV and banana bract mosaic virus (BBrMV)). The particles were approximately 580 nm in size, suggestive of members of the family Betaflexiviridae. Like all virions of the latter family, these particles showed a diameter ranging from 10 to 15 nm [33].

### 3.2. High-Throughput Sequencing and Bioinformatics Analysis

The HTS yielded a total number of 8,683,460 sequence reads ranging from 50 to 151 nucleotides. After removing duplicates and de novo assembly, a total of 101,160 contigs was obtained, summing 33,003,893 nucleotides. The maximum, minimum and mean lengths were 16,883 bp, 86 bp and 386 bp, respectively. Two contigs, contig1 and contig 2, each approximately 5 kb in length, had high homology (E-value of 0) to the reference *Banana mild mosaic virus* (BanMMV) sequence (GenBank accession NC_002729). The contigs were further extended by iterative mapping to lengths of 7336 and 7311 nt, respectively. The genome coverage of both isolates reached almost 100%.

The two new nucleotide sequences were deposited on GenBank. They are hereafter called MT872724 and MT872725 with their GenBank accessions numbers, respectively, for contig 1 and contig 2. They shared a pairwise identity of 75.3% at the nucleotide level, and had pairwise identities of 75.2% and 75.4%, respectively, with the BanMMV reference genome (GenBank accession NC_002729). At the amino acid level, MT872724 and MT872725 shared a pairwise identity of 40%. They had pairwise identities of 76.5% and 75.9%, respectively, with the translated amino acid sequence of the BanMMV reference genome (GenBank accession NC_002729). MT872724 and MT872725 shared 80% nt identity (91.1% aa identity) between their respective CPs. They shared 76% nt identity (84.3% aa identity) between their respective RdRps. All percentages of identity between BanMMV (GenBank accession NC_002729) and the two isolates regarding CP and polymerase genes, either at the nt or the aa level, are detailed in Table 1.

Throughout the Betaflexiviridae family, isolates of different species should have less than about 72% nt identity (or 80% aa identity) between their respective CP and polymerase genes [34]. Therefore, MT872724 and MT872725 can be considered as different isolates of BanMMV.

### 3.3. Phylogenetic Analysis

The whole genome sequences of MT872724 and MT872725 (7336 nt and 7311 nt, respectively) were compared and aligned together with the single complete genome of BanMMV (GenBank accession NC_002729; 7352 nt), and with other members from the different genera of the Betaflexiviridae family. The phylogenetic tree reconstructed from the full genome sequences (Figure 2) showed the clustering of the new genomes into a group along with BanMMV, confirming their taxonomical placement as novel isolates of this unassigned member of the Betaflexiviridae family. They are also close to the other genera of the same family, but the closest one seems to be the Foveavirus, as it is the genus of Peach chlorotic mottle virus.

The phylogenetic tree reconstructed from CP nucleotide sequences (Appendix A) had a similar topology to the one inferred from full genomes, and also showed that the new genomes clustered together into a group with the single BanMMV genome (GenBank accession NC_002729). Further analysis was carried out in order to assess how the new genomes of BanMMV related to the currently known viral diversity. Phylogenetic analysis inferred from RdRp nucleotide sequences showed that several monophyletic groups were generated, which confirms that BanMMV is a genetically diverse virus species (Figure 3). Similarly, Meena et al. [35] reported the extreme variability of the RdRp genomic region of flexiviruses. In the case of the novel genomes characterized during this investigation, each genome clustered in a different group.

The co-existence of two distinct and divergent isolates of BanMMV in the same plant can be explained by the possibility of the horizontal transfer of the virus between plants. Similarly, a previous analysis supported the existence of the horizontal spread of BanMMV between plants [8]. In fact, Caruana and Galzi [36] showed that virus-free planting material became infected with CMV and BanMMV once planted in the field. In the same line, Teycheney et al. [3] reported that identical RdRp sequences were obtained from two independent pairs of plants, and there was a strong dispersion gradient of the virus between plants. No insight into the mechanism underlying the plant to plant transfer has yet been provided. Mechanisms are still unknown. Subsequently, further studies need to be done [3,4,8].

The percentage identities of RdRp and CP nucleic acid and the translated amino acid sequences confirmed that both contigs are independent full genomes of BanMMV [33].

### 3.4. Genetic Diversity of the Two Identified Isolates

The analysis of minor SNPs in the two novel BanMMV genomes resulted in 31 SNPs for MT872724 (of which 2 are on the non-coding region) and 14 SNPs for MT872725 (Table 2). Details about base change, variant frequency and SNP type are shown in Appendix A.

The number of SNPs in the RdRp varied from 9 to 20 between the two BanMMV viral populations from this study; there were one to two SNPs in TGB2, one to three SNPs in TGB3, and one to three SNPs in the CP region. No SNP was detected in the TGB4 region for MT872725, whereas two SNPs were located in the same region for MT872724.

The percentages of SNPs in the five regions are 4.9, 4.6, 3.3, 4.5 and 5.5%, respectively for RdRp, TGB2, TGB3, TGB4 and CP. The variant frequency was slightly higher in the CP coding region. This is in line with a previous study [3], which showed that little selection pressure is applied on BanMMV coding genomic sequences above that necessary for the conservation of the encoded proteins.

Around 80% of SNPs for MT872724 and the majority of SNPs for MT872725 were present at a frequency of less than or equal to 10%, supposing that large numbers of low frequency variants are present in sequence clouds of BanMMV genotypes [37]. These minor SNPs support the fact that the quasispecies is a well-connected group of variants, and not a collection of random mutants. In the same line, [38] such a cloud of variants was proposed to be genetically linked through mutation, to interact cooperatively on a functional level, and to collectively contribute to the characteristics of the population. For the same isolate, 5 out the 31 SNPs had a high frequency greater than 20%, suggesting that they might be part of mutant networks [37]. No SNPs located at the same nucleotide position have been identified in both quasispecies.

### 3.5. Nucleotide Sequence Analysis and Primer Design

The 210 available nucleotide sequences of the BanMMV isolates were downloaded from NCBI and aligned together with complete genomes of MT872724 and MT872725. Around half of the bases covered by several sequences presented a polymorphism, which makes primer design more complicated. In this context, a previous study described that fairly high nucleotide sequence variation in conserved regions of the polymerase and coat protein has been reported between isolates, which is frequently observed for this family of viruses [39]. In the same vein, Teycheney et al. [3] supposed through an observation of similar levels of amino acid conservation between pairs of isolates that there were wide variations in their nucleotide sequences. Likewise, high variabilities were observed for other members of the Betaflexiviridae family, which suggests that this property is shared by a large number of members of this family. The alignment results of all BanMMV published primers with the two novel genomes are shown with details in Appendix A. In fact, the two following primer pairs, BanCP1 and BanCP2 [3] and BanMMVCPFP and BanMMVCPRP [40], were unable (based on in silico tests) to bind with MT872724 and MT872725. Thus, no molecular detection could be achieved with these primers.

For PDO-F2i and PDO-R1i [15], there was binding with both isolates. However, one mismatch in the 3′ end was detected for each primer, in both cases. A molecular test with the primer pair was not done since the nested PCR method generates highly concentrated amplicons and a high risk of contamination [41]. This might be a serious constraint, especially for routine diagnostic centers.

In addition to these three primer pairs, there are three other forward primers that could bind (with Poty1) to BanMMV isolates on the CP region. Both molecular and in silico tests are available for these primers (Appendix A). The three pairs are BanMMCP2 and Poty1, BanMMV CP8 and Poty1, and BanMMV CP9 and Poty1.In the case of BanMMCP2, thre mismatches were present in the 10 bases of the 3′ end (including one mismatch in the 5 bases of the 3′ end) for MT872724, and two mismatches were detected in the 10 bases of the 3′ end (no mismatch detected in the 5 bases of the 3′ end) for MT872725. Five mismatches were detected in the 25 bases of the primer for both genomes. This could explain the absence of molecular detection with the primers used for the routine indexing of BanMMV. In the case of BanMMV CP8, although there are no reported mismatches in the 10 bases of the 3′ end of BanMMV CP8, as they are covered by the degenerate bases (but three mismatches the whole primer length), the primer failed to give amplicons with ITC0763. This might be explained by the presence of many degenerate bases (at least four degenerate bases) in the 3′ end of the primer. In fact, it has been reported that primers with high levels of degeneracy could decrease the specific detection of the virus [42].

As such, there is a need to develop a polyvalent and degenerated primer with inosine bases to detect a broad range of BanMMV isolates. In this context, a previous survey reported the advantages of the use of Inosine-containing primers [43]. In fact, this kind of primer is considered very useful in detecting more diverse populations in the environment. The use of inosine enhances primer inclusiveness while retaining exclusiveness. In the same line as the current survey, inosine is the best base to use in particular sites with three or four base ambiguities [43,44].

A new primer, called BanMMV CP9, was designed to amplify, with Poty-1, a fragment of BanMMV coat protein. In silico evaluation has been carried out in order to highlight potential mismatches for the 54 CP sequences of BanMMV available on GenBank, the single complete genome of the virus and the new isolates from this study. Data are shown in Appendix A. According to these results, BanMMV CP9 could be able to detect all these isolates since it presents only one mismatch at the fourth (JX183725.1) or fifth base (AY730743.1) in the 3′ end of the primer.

### 3.6. Primer Validation

Based on the results of gradient PCR using the new primer, 51.6 °C was considered as the optimal annealing temperature (Ta) to proceed with the PCR assay with BanMMV CP9. The concentration of 20 µM for this primer has been kept since other concentrations tested did not improve the results. As expected, the newly developed primer was able to detect BanMMV from our sample of interest (Appendix A), and from other samples tested with this primer.

Further analysis was carried out in order to confirm that the obtained amplicons came from the amplification of BanMMV sequences, and not from other origins. As is shown in Table 3, BanMMV genomes detected by RT-PCR using BanMMV CP9, then sanger-sequenced, exhibited considerable homology to multiple BanMMV sequences. This homology varied from 69% to 89% with BanMMV CP sequences of different isolates.

### 3.7. Laboratory Validation of the New Primer in Banana Germplasm Collection

In order to validate the primer, a retrospective analysis of 110 accessions from different geographical origins and with different viral statuses was conducted, comparing BanMMCP2 and BanMMV CP9 forward primers as an IC-RT-PCR test with the Poty1 reverse primer. These PCR tests were carried out on the complementary DNAs (cDNAs) prepared with Poty-1 from 110 accessions. Detailed results are shown in Appendix A. Of these 110 accessions, 105 (95.45%) showed a concordance in the results with both primers (16 positive accessions and 89 negative accessions), while five discrepancies were noticed. Previously undetected infections of two accessions were identified only by BanMMV CP9. Three infections were missed by the same primer but detected by BanMMCP2. The five detections were confirmed by sequencing of the PCR product.

The three BanMMV-infected accessions that were not recognized by BanMMV CP9 have different origins—ITC1709 originates from Cameroon, ITC1894 from China and, unfortunately, the origin of ITC1686 is unknown. This downplays the hypothesis that the undetected isolates correspond exclusively to a certain geographic origin. This loss of inclusiveness could be explained by the decrease in its robustness due to the presence of either three degenerate bases or two inosine bases. In the same context, Ref. [45] confirmed that universal primers containing inosine do not necessarily amplify all heterologous or divergent sequences of different species of viruses.

In a preliminary analysis, the analytical sensitivity of the new diagnostic primer (BanMMV CP9) was similar to, and even better than, that of the CP2 primer on three accessions. The results were identical for two accessions, but, for one accession, CP9 was still able to detect one isolate of the virus with a 100× diluted sample, while BanMMV CP2 failed. Details are shown in Appendix A.

## 4. Conclusions

Viral diseases, among which is BanMMV, are one of the main constraints for *Musa* germplasm movement and vegetative propagation. Sensitive and specific diagnostic tools are needed to help in their control. It is nevertheless highly challenging to develop inclusive primers that could detect all BanMMV isolates, which is supported by several studies reporting the very high level of molecular variability of this virus and other species from the Betaflexiviridae family [13]. So far, six primer pairs have been designed and published to detect this virus, but in silico analyses, completed by laboratory tests for some primer pairs, suggested that all the BanMMV isolates could not be detected by a single test because of the presence of several mismatches, particularly at the 3′ end of primers.

In the current study, two divergent isolates of this virus were fully sequenced, adding two full genomes of BanMMV to the unique genome in GenBank. A new degenerate, inosine-containing diagnostic primer was designed to improve the inclusiveness of the detection protocol, which failed to detect these novel isolates. In fact, this kind of primer could help to overcome the high genetic diversity of the virus, even if a lower sensitivity could sometimes result from the degeneracy and inosine contents of the primers [9,13,14]. The new primer detected BanMMV infection in two samples that tested negative with the existing primer, but failed to detect three infections among the 110 tested accessions.

Therefore, as the very high genetic diversity of BanMMV is a strong challenge for the development of a single protocol with appropriate inclusiveness, our recommended approach would be to continue to use one of these primers for routine indexing and to confirm negative results with the other primer. This two-step strategy might be preferable over the use of nested PCR, which would require two PCR reactions for all samples, with a high risk of contamination.

On the other hand, this can be time-consuming, and unknown divergent isolates might exist. As such, HTS technologies that analyze the plant virome present in a sample with minimal bias and independently of the genome sequence of the target can be a good alternative for routine indexing in the future.

## Figures and Tables

**Figure 1 pathogens-09-01045-f001:**
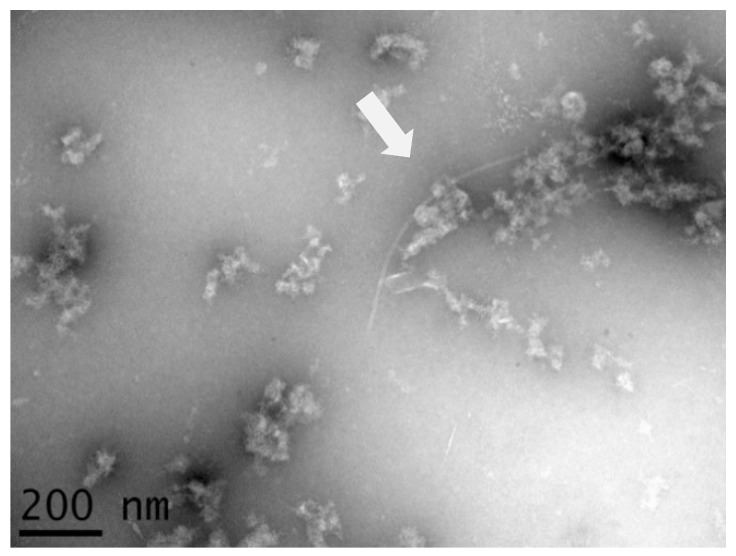
Flexuous filamentous particle (arrow) obtained after viral particle enrichment from leaves of banana accession ITC0763. The scale bar represents 200 nm.

**Figure 2 pathogens-09-01045-f002:**
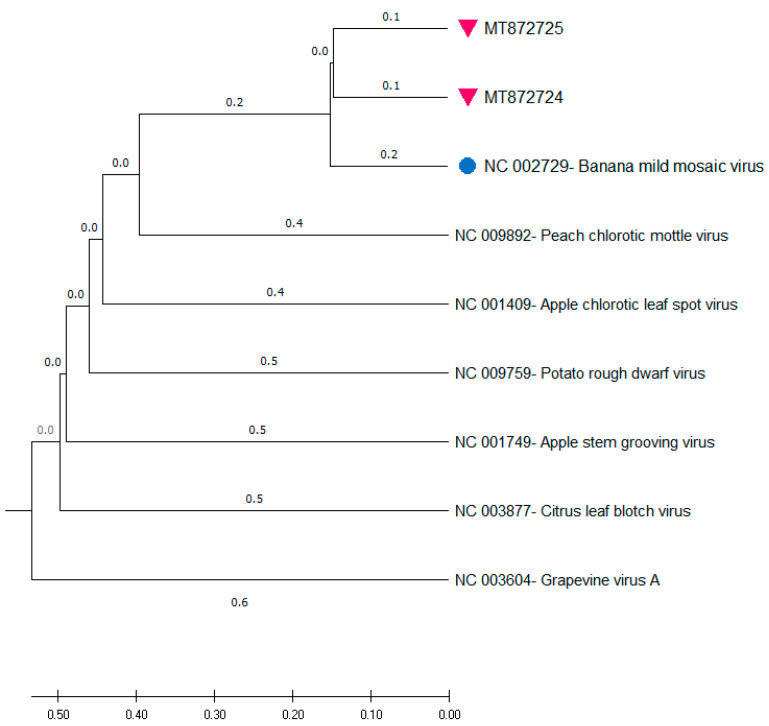
Unweighted Pair Group Method with Arithmic mean (UPGMA) phylogenetic tree inferred from full genomes of nt sequences of BanMMV, isolates from this study, and members from different genera of the Betaflexiviridae family. The cyan dot denotes the single complete genome of the virus (GenBank accession NC_002729). The purple triangles denote the BanMMV isolates characterized during this study. Branches were bootstrapped with 1000 replications. The scale bar indicates the number of substitutions per site.

**Figure 3 pathogens-09-01045-f003:**
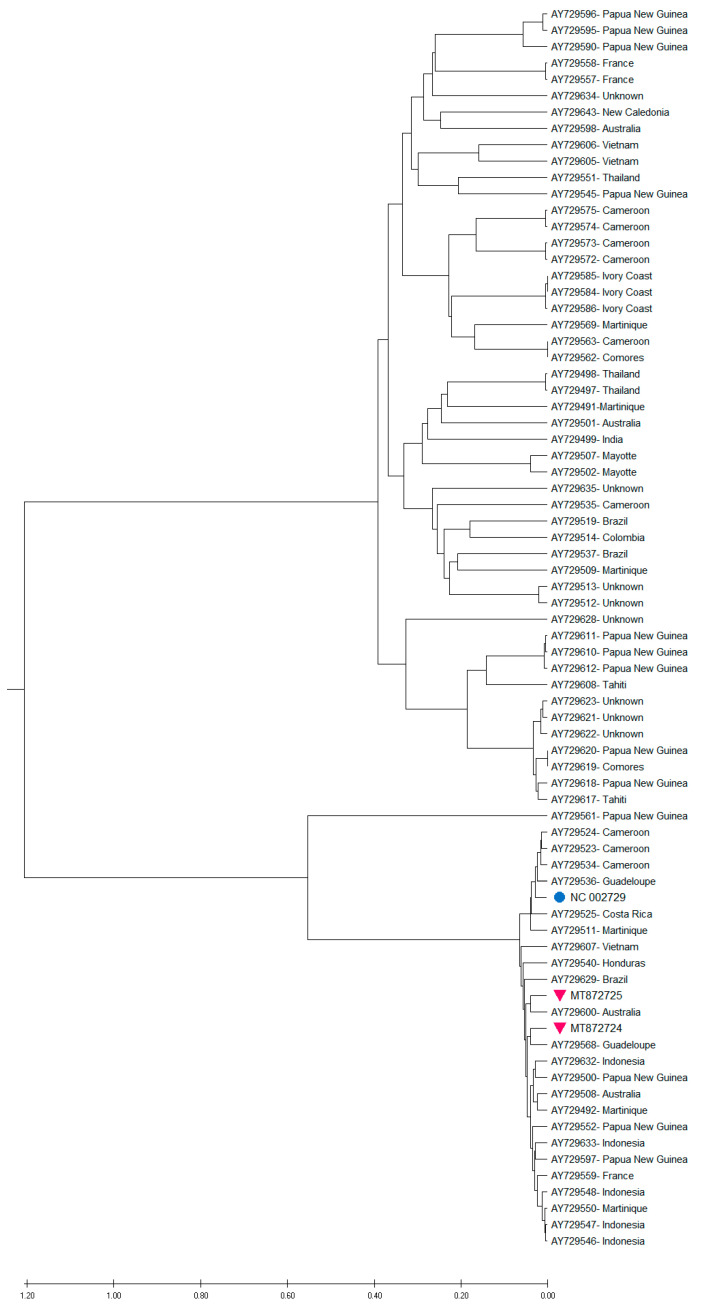
UPGMA phylogenetic tree inferred from the RdRp aa sequences of BanMMV and isolates from this study. The cyan dot denotes the single complete genome of the virus (GenBank accession NC_002729). The purple triangles denote the BanMMV isolates characterized during this study. Branches were bootstrapped with 1000 replications. The scale bar indicates the number of substitutions per site.

**Table 1 pathogens-09-01045-t001:** Percentage of identity between BanMMV reference genome and the two novel isolates regarding CP and polymerase genes, at both the nucleotide (A) and amino acid (B) levels.

		(A) Nucleotide Level				(B) Amino Acid Level			
		MT872724		MT872725		MT872724		MT872725	
		CP	RdRp	CP	RdRp	CP	RdRp	CP	RdRp
BanMMV NC_002729	CP	77.6%	n.a.	76.8%	n.a.	88%	n.a.	87.8%	n.a.
	RdRpp	n.a.	75.4%	n.a.	75.4%	n.a.	81%	n.a.	81.6%

With n.a. refers to non-available.

**Table 2 pathogens-09-01045-t002:** Distribution of single nucleotide polymorphisms (SNPs) with a frequency higher than 1% in the five ORFs of BanMMV populations.

**A- First Novel BanMMV Genome MT872724**						
**BanMMV New Genome 1**	**Total SNPs**	**RdRp**	**TGB2**	**TGB3**	**TGB4**	**CP**
MT872724	31	20	2	1	2	3
Length (nt)	-	5328	675	339	213	678
**B- Second Novel BanMMV Genome MT872725**						
**BanMMV New Genome 2**	**Total SNPs**	**RdRp**	**TGB2**	**TGB3**	**TGB4**	**CP**
MT872725	14	9	1	3	0	1
Length (nt)	-	5307	675	339	213	717

With RdRp: RNA-dependent RNA polymerase; TGB: triple gene block; CP: coat protein.

**Table 3 pathogens-09-01045-t003:** BlastN results of sequenced samples tested with BanMMV CP9.

Sample	BlastN Results		
	% nt Identity	Sequence ID	«Organism»
ITC 1022	84%	AY730737.1	BanMMV isolate CP2.3 CP gene, partial cds
ITC 1709	89%	AY730742.1	BanMMV isolate CP4.2 CP gene, partial cds
ITC 1746	84%	KT780866.1	BanMMV isolate TN1 CP gene, complete cds
ITC 1518	85%	AY730732.1	BanMMV isolate CP18.1 CP gene, partial cds
ITC 1628	87%	AY730744.1	BanMMV isolate CP6.1 CP gene, partial cds
ITC 1651	83%	AY730748.1	BanMMV isolate CP77.2 CP gene, partial cds
ITC1681	87%	AY730743.1	BanMMV isolate CP4.3 CP gene, partial cds
ITC 1733	81%	AY730754.1	BanMMV isolate CP82.2 CP gene, partial cds
ITC 1740	83%	AY730740.1	BanMMV isolate CP3.3 CP gene, partial cds
ITC 1794	77%	AY366187.1	BanMMV isolate F8 CP gene, complete cds
ITC 1929	69%	EF143978.1	BanMMV isolate Q6.2T CP gene, complete cds

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
