# Peer review of "Identification of Divergent Isolates of Banana Mild Mosaic Virus and Development of a New Diagnostic Primer to Improve Detection"

_pathogens, 2020, doi:10.3390/pathogens9121045_

Round 1

Reviewer 1 Report

This study presesnts two reliable primers to detect Banana mild mosaic virus (BanMMV) using high-throughput sequencing (HTS) on 110 accessions from the international banana germplasm collection. To generate optimal sets of primers, the authors used HTS and obtained two complete genomes of BanMMV. The results were pretty well organized based on things that authors found. The hypothesis is clear, experimental design is proper, the results and conclusions were convincing. The work has been carried out in a comprehensive fashion and is ready for publication.

Author Response

Reviewer1:

This study presents two reliable primers to detect Banana mild mosaic virus (BanMMV) using high-throughput sequencing (HTS) on 110 accessions from the international banana germplasm collection. To generate optimal sets of primers, the authors used HTS and obtained two complete genomes of BanMMV. The results were pretty well organized based on things that authors found. The hypothesis is clear, experimental design is proper, the results and conclusions were convincing. The work has been carried out in a comprehensive fashion and is ready for publication.

We thank you very much for your positive and encouraging feedback.

Sincerely,

Reviewer 2 Report

The paper describes the challenges in designing PCR primers to detect a group of viruses with high diversity between isolates. Paper is well written, but it requires minor changes in the English language. Below I list several questions and comments about the manuscript that, in my view, will improve it. I recommend Authors to address them as best as they can.

Title: Title does not reflect the work being presented in the paper. The main objective was to design a new PCR primer to improve diagnostics. Title should reflect this and not focus on “phylogenomics” or diversity revealed by HTS. HTS was used as a tool to design the new primer.

Abstract:

Lane 18: what are “inclusive primers”? Please edit this sentence.

Lane 33: “ensuring analytical sensitivity”. High or low sensitivity? There are no results to support this. Please remove or add more results to support this sentence.

Introduction:

Lane 44: “healthy germplasm”:  testing germplasm for viruses only guarantees “virus-free germplasm”. Please edit this sentence.

Lane 48: change “plant pests” to “plant pathogens”

Lane 49: add “of planting material” at the end of the sentence.

Lane 62: “transitory symptoms”: please add which ones?

Lanes 80-83: This needs rephrasing and clarification. Monoclonal antibodies are very specific so can lead to false negative results; polyclonal antibodies if picking plant material how can they be used efficiently in ISEM and/or IC?

Lanes 90-92: there are good examples of primers with high levels of degeneracy used in diagnostics. For example, the generic badnavirus primer pair (Yang et al, Arch. Virol. 2003, 148, 1957–1968).

Lane 102: “GenBank sequences”: Authors need to be more specific which sequences were used?

Include more information on what the “new diagnostic primer” are used for?

Materials and Methods:

Section 2.1

Lanes 107-111: Move this information to results section. BBrMV needs definition. All of the section 2.1 needs more clarification. How many plants were grown in the greenhouse? How many were tested? For accession ITC0763, did you sample 3 leaves and 8 discs/leaf? This makes 24 samples. All 24 samples were tested for 1 accession?

Lane 117: the 110 accessions were treated as ITC0763. Does this mean that 24 samples/accession = 2640 samples in total?! Please clarify this.

Section 2.3

Lane 126: Please describe what is the “CEB buffer”

Section 2.4

More detail is needed in this section! What molecular diagnostic tests? Did you only use crude extracts as templates? No RNAs were used? What is “Poty-1” and what protocol was used? There are no results about the use of primer BanMMV CP8!

Did you use primers targeting an internal control gene to avoid false-negative results? If yes please include it in the results section.

Section 2.5

How many samples were tested by ISEM? What is the source of the antibodies?

Section 2.7

Lane 149: Why the use of the ref-seq database and not the complete NCBI viral database?

Lane 155: whole genome alignments are not shown on the results!

Lane 156: Please give more information on SNP analysis.

Section 2.8

Lane 162: Include link to the ORF finder tool

Lane 163: what is the “selection of RdRP nucleotide sequences”? Please clarify.

Lane 164: Please add reference for MEGA software. The sentence needs rephasing. As it is, it seems that phylogenetic analysis was done with complete and RdRP sequences together, which I guess was not what Authors did.

Lane 171: typo in “tress”. Please change to “trees”.

Section 2.9

Lane 176: “A single complete genome of BanMMV was available”. what about the 2 new isolates obtained in this study? Please edit this sentence.

Lane 183: “Suppl. file S2a”. Please include the location of primers on the reference sequence.  

Section 2.10

Lane 186: Healthy accessions are not described in section 2.1. Please add this information.

Results and Discussion

All supplementary files need better and detailed legends and titles.

Lane 190-191: ISEM used in the study is a specific approach as antibodies are used for coating and for that one must know the target. Please edit this sentence.

Lane 192: How can electron microscopy detect sequence variants?

Lanes 205-208: BanMMV genome is about 7.3kb. If Authors obtained a 5kb contig how can the genome coverage be 100%? Please revise this section.

Lane 215: identity percentages are different from the ones shown on lane 206. This is confusing. Maybe delete the percentage info on lane 206? It needs clarification.

Lane 216: change “protein” to amino acid level. Why not include results of percentage identity on the CP and polymerase as per ICTV recommendation?

Lane 228: Figure 2. This figure does not bring new information. And does not show bootstrap values. Please delete or re-do a phylogenetic analysis including members of different genus of the family Betaflexiviridae. You could prepare trees with full genomes and amino acid sequences of the replicase.

Lane 231: “in the same vein”: please change to “similar topology”

Lane 242: Please explain further what you mean by “plant-to-plant transfer”. The paragraph is confusing. Also, why not discuss the possibility of recombination? You could do a quick recombination analysis and include more information on the results and discussion.

Lane 250: Figure 3 does not show bootstrap values. Same for the Suppl. file S3.

Lanes 262-263: where is “a” and “b” on the table?

Lane 274: “CP coding region more varialble”. This is not reflected on the trees. According to RdRp and CP trees, isolates seem more distant on the RdRp region. This needs clarification.

Lane 278: “almost all SNPs” – this is vague. Please rephrase.

Lane 290-291: Please re-do this sentence.

Lanes 292-298: “28%”. This percentage is the ICTV recommendation threshold to assign different species within the family Betaflexiviridae. This paragraph needs rephrasing because is confusing and I am not sure what is the information you are giving.

Lane 304: “Suppl. file S2-b”. This table is confusing. For example, binding to what sequence? And how can a BanCP1-BanCP2 have 3,127 gaps?? Please make table clearer. Why only do in silico tests and not try all these primers in the lab to get solid results?

Lane 317: “BanMMV CP8 primer”. this primer is the same as BanMMV CP2 but with degenerate bases. As so, there are no reported mismatches on this primer because they are covered by the degenerate bases. Please edit this information on the manuscript.

Lane 319: Primer BanMMV CP9 has 3 degenerate bases in the 3’ end including a “I”. Therefore, the explanation given that the number of degenerate bases on primer BanMMV CP8 is causing the negative results needs rethinking.

Lane 337: “Suppl. file S6”. primer-dimers are as strong as the expected fragment. Further optimization is required on the PCR. Why did you just do a gradient PCR and did not optimise the PCR conditions such as magnesium and primers concentrations?

Lane 341: the percentages are related to the 280bp PCR fragment which is less than half of the full-length CP. As so, I think these results are not significant and should be removed unless you show that the 280 bp region of the CP is a representation of the full CP.

Lane 346: why not also test BanMMV CP8 as this primer seems an optimised version of BanMMV CP2?

Lane 348: How were the 110 accessions previously tested? which method was used?

Lane 350-352: But 3 of those accessions gave positive results only with BanMMV CP2. One more positive than with BanMMV CP9. How can you be sure if primers sensitivity is the same or different if this was not tested? Or at least not shown?

Lane 358: Where is the data to support the specificity and sensitivity of the primer?

Lanes 360-365: this paragraph clearly contradicts the information on lanes 321-226. A better justification is needed for the limitations of primer BanMMV CP9.

Lane 366: How can the primers be used in “complementary”? If you need to use 2 primers for every sample the costs will increase substantially. This does not seem ideal.

Lane 367: Again, why not testing the sensitivity of the primers instead of predicting with no data to support this?

Lane 371: Another contradiction! here you say inosine can overcome high genetic diversity but on lane 362 you say that inosine do not amplify all divergent sequences!!

Conclusions

Lane 378: “important economic losses”. this is trying to bring more importance to BanMMV than it has. According to the European Food Safety Authority, BanMMV is not associated with significant yield losses in banana (EFSA Journal 2008 652,1-21). Please rephrase this.

Lane 386: There is no data to support mutation rates. Pease delete this.

Lane 389: most tests were done in silico. Why didn’t you analyse (in the lab) the samples used in this study with all published primers described on suppl. file S2? It could give a much better comparison of which primer set is the best.

Lane 393: there are many contradictions throughout the manuscript about the utility of the inosine. This needs to be corrected. I think the new primer BanMMV CP9 does not improve the diagnostics of BanMMV and you’re trying to justifying it. If inosine can overcome the high genetic diversity of the virus why did it failed to detect 3 isolates that were positive with primer BanMMV CP2.

Lane 398: this will increase the costs/test. Do you think is the best way to perform diagnostics for BanMMV?

Lane 406: I agree you need to design a better primer that detects all diversity. the BanMMV CP9 does not solve your problems and in my view does not improve diagnostics.

Author Response

Reviewer2:

The paper describes the challenges in designing PCR primers to detect a group of viruses with high diversity between isolates. Paper is well written, but it requires minor changes in the English language. Below I list several questions and comments about the manuscript that, in my view, will improve it. I recommend Authors to address them as best as they can.

We would like to thank you for your feedback and your highly constructive remarks. In fact, your critical thinking has given us a lot of ideas about the perspectives and about how the work could be improved. All your comments were treated, point by point. Please find hereunder all the answers to your suggestions and remarks. All modifications were highlighted In yellow in the whole manuscript.

Title: Title does not reflect the work being presented in the paper. The main objective was to design a new PCR primer to improve diagnostics. Title should reflect this and not focus on “phylogenomics” or diversity revealed by HTS. HTS was used as a tool to design the new primer.

Title has been completely changed and improved in order to reflect better the main objective of the work.

Abstract:

Lane 18: what are “inclusive primers”? Please edit this sentence. Lane 18: « Inclusive primers » was replaced by specific primers.

Lane 33: “ensuring analytical sensitivity”. High or low sensitivity? There are no results to support this. Please remove or add more results to support this sentence. Sentence removed.

Introduction:

Lane 44: “healthy germplasm”:  testing germplasm for viruses only guarantees “virus-free germplasm”. Please edit this sentence. Lane 44 : the adjective was removed.

Lane 102: “GenBank sequences”: Authors need to be more specific which sequences were used? Lane 102 : the sentence was rephrased as following  « … based on the virus sequences available in GenBank… ».

Materials and Methods:

Section 2.1

Lanes 107-111: Move this information to results section. BBrMV needs definition. All of the section 2.1 needs more clarification. How many plants were grown in the greenhouse? How many were tested? For accession ITC0763, did you sample 3 leaves and 8 discs/leaf? This makes 24 samples. All 24 samples were tested for 1 accession?  Lane 117: the 110 accessions were treated as ITC0763. Does this mean that 24 samples/accession = 2640 samples in total?! Please clarify this.

 Information mentioned has been removed to results subsection 3.1. All the section 2.1. was improved and more clarified. The virus name has been spelled out.  Yes the 110 accessions were treated as ITC0763. This does not mean 2640 samples, we do not count like that.

Section 2.3

Lane 126: Please describe what is the “CEB buffer”  

The composition of CEB buffer has been added in subsection 2.3.

Section 2.4

More detail is needed in this section! What molecular diagnostic tests? Did you only use crude extracts as templates? No RNAs were used? What is “Poty-1” and what protocol was used? There are no results about the use of primer BanMMV CP8!

All the details about molecular diagnostic test have been added in a new supplementary file. We have also mentioned in subsection 2.4 that the PCR tests were carried out on the cdnas produced after an IC-RT assay (from the crudes extracts) to give more clarification to the section.

Did you use primers targeting an internal control gene to avoid false-negative results? If yes please include it in the results section. No primers targeting an internal control gene was used to avoid false-negative results.

 Section 2.5

How many samples were tested by ISEM? What is the source of the antibodies?

We have added a supplementary file describing in details all the procedure of ISEM (including the source of antibodies). To answer to your question, 3-5 clonal plants of a germplasm line were grown out from tissue culture under glasshouse conditions. Plants were sampled at three and six months after deflasking. Samples at each time point were pooled for the indexing process. Unidentified flexuous rod-shaped particles were detected in accession/line ITC0763 at both three- and six-month sampling times.

Section 2.7

Lane 149: Why the use of the ref-seq database and not the complete NCBI viral database? This was especially for rapidity and non-redundancy. Indeed, results are the same with both bases.

Lane 155: whole genome alignments are not shown on the results! Since we have around 200 sequences of the virus (including the 2 new genomes), it is really hard to represent the alignments of all these sequences. This cannot be included in the manuscript. If needed, a supplementary figure can be added.

Lane 156: Please give more information on SNP analysis. More information has been added in this section concerning the parameters of SNP calling.

 Section 2.8

Lane 162: Include link to the ORF finder tool. Link to the ORF finder toll added.

Lane 163: what is the “selection of RdRP nucleotide sequences”? Please clarify. Corrected.

Lane 164: Please add reference for MEGA software. The sentence needs rephasing. As it is, it seems that phylogenetic analysis was done with complete and RdRP sequences together, which I guess was not what Authors did. The whole subsection has been edited and information was clarified and added.

Lane 171: typo in “tress”. Please change to “trees”. Corrected.

 Section 2.9

Lane 176: “A single complete genome of BanMMV was available”. what about the 2 new isolates obtained in this study? Please edit this sentence.

It was mentioned « a single complete genome “ because we were talking about the sequences available in the GenBank database before the identification of the genomes from this study. Therefore, we mentioned in the sentence just below that the new primer was designed on these sequences but also using the new genomes of this study.

Lane 183: “Suppl. file S2a”. Please include the location of primers on the reference sequence.  

A location column has been included in the supplementary file relative to published BanMMV primers.

Section 2.10

Lane 186: Healthy accessions are not described in section 2.1. Please add this information.

The information was also added in subsection 2.1.       

Results and Discussion:

All supplementary files need better and detailed legends and titles. Done.

Lane 190-191: ISEM used in the study is a specific approach as antibodies are used for coating and for that one must know the target. Please edit this sentence. Done. Sentence edited as following : “Like PCR tests and other methods in plant virus diagnostics, electron microscopy is a specific approach that is based on the use of antibodies and requires previous knowledge of the expected targets”.

Lane 192: How can electron microscopy detect sequence variants? The sentence was edited to bring more clarification.

Lane 228: Figure 2. This figure does not bring new information. And does not show bootstrap values. Please delete or re-do a phylogenetic analysis including members of different genus of the family Betaflexiviridae. You could prepare trees with full genomes and amino acid sequences of the replicase. This phylogenetic analysis was reconducted using nucleotide sequences of the new genomes, BanMMV, and other members from the different genera of the family.

Lane 231: “in the same vein”: please change to “similar topology” Ok. Done.

Lane 242: Please explain further what you mean by “plant-to-plant transfer”. The paragraph is confusing. Also, why not discuss the possibility of recombination? You could do a quick recombination analysis and include more information on the results and discussion.

We mean horizontal transfer between plants (it has been corrected in the manuscript). The availability of only two full genome (+ 1 from NCBI) makes the recombination analysis not so robust (the more full genome included, the more reliable would be the recombination analysis). So, we did not do the recombination analysis to continue focusing the publication on the development of diagnostic primers.

We really appreciate your suggestion. It will be considered in future works when more genome sequences are available.

Lane 250: Figure 3 does not show bootstrap values. Same for the Suppl. file S3. Done.

Lanes 262-263: where is “a” and “b” on the table?

A & B were used in order to show that there are two parts in the Table 1 (becoming Table 2 after revision)  (Table 2-A, Table 2-B). We have modified it to bring more clarification.

Lane 274: “CP coding region more varialble”. This is not reflected on the trees. According to RdRp and CP trees, isolates seem more distant on the RdRp region. This needs clarification.

It is indeed much more clarified with the new RdRp aa sequences added. This latter confirmed the idea.

Lane 278: “almost all SNPs” – this is vague. Please rephrase. Done.

Lane 290-291: Please re-do this sentence. This is done.

Lanes 292-298: “28%”. This percentage is the ICTV recommendation threshold to assign different species within the family Betaflexiviridae. This paragraph needs rephrasing because is confusing and I am not sure what is the information you are giving. Done.

Lane 304: “Suppl. file S2-b”. This table is confusing. For example, binding to what sequence? And how can a BanCP1-BanCP2 have 3,127 gaps?? Please make table clearer. Why only do in silico tests and not try all these primers in the lab to get solid results?

Thank you very much for your comment. The analysis of matches between all published BanMMV primers and the novel isolates was reconducted using the “test with saved primers” menu in Geneious. In fact, there were some errors (especially the gaps). Everything was edited. Modifications were also added regarding the interpretation of this part of results in the manuscript.

The problem with some primers that there was no binding with in silico tests. So, they were discarded for PCR tests (BanCP1-BanCP2 and BanMMVCPFP-BanMMVCPRP). All details are now added in results & discussion part.

Lane 317: “BanMMV CP8 primer”. this primer is the same as BanMMV CP2 but with degenerate bases. As so, there are no reported mismatches on this primer because they are covered by the degenerate bases. Please edit this information on the manuscript. The sentence was rephrased and edited as required.

Lane 319: Primer BanMMV CP9 has 3 degenerate bases in the 3’ end including a “I”. Therefore, the explanation given that the number of degenerate bases on primer BanMMV CP8 is causing the negative results needs rethinking. Thank you very much. The degenerate bases in the two primers are not the same. Therefore, BanMMV CP8 has 6 degenerate bases whereas BanMMV CP9 has 3 degenerate bases and two inosine bases. In 3’ end, BanMMV CP8 has more degenerate bases than BanMMV CP9. .

Lane 337: “Suppl. file S6”. primer-dimers are as strong as the expected fragment. Further optimization is required on the PCR. Why did you just do a gradient PCR and did not optimise the PCR conditions such as magnesium and primers concentrations?

Thank you for your comment. Indeed the primer dimers are strong and might be also caused by the use of inosine and other degenerated bases. We have also tried with other primer concentrations (e.g: 25 µM) without improvement of the results.

Lane 341: the percentages are related to the 280bp PCR fragment which is less than half of the full-length CP. As so, I think these results are not significant and should be removed unless you show that the 280 bp region of the CP is a representation of the full CP. The 280 bp has been removed as required.

Lane 348: How were the 110 accessions previously tested? which method was used? Crude extracts were prepared as described in the MM section. An IC-RT assay using Poty-1 was carried out from these 110 samples. Thus, 110 cDNAs were produced. PCR tests were performed on these cDNAs with either CP2 or CP9. So PCRs with different primers were performed on the same cDNAs.

Lane 350-352: But 3 of those accessions gave positive results only with BanMMV CP2. One more positive than with BanMMV CP9. How can you be sure if primers sensitivity is the same or different if this was not tested? Or at least not shown?

Lane 358: Where is the data to support the specificity and sensitivity of the primer? Please see response to comment above.

Lanes 360-365: this paragraph clearly contradicts the information on lanes 321-226. A better justification is needed for the limitations of primer BanMMV CP9. The part of manuscript that presents contraction has been removed.

Lane 366: How can the primers be used in “complementary”? If you need to use 2 primers for every sample the costs will increase substantially. This does not seem ideal.

Thank you for your comment. The BanMMV virus is known to display a very high genetic diversity. The development of a single protocol to detect all the virus isolates is highly challenging.

Through this study, we recommended the use of one of the two following primers (BanMMCP2 and BanMMV CP9; the new primer) for routine indexing, and to confirm negative results with the other primer. This technique could be time-consuming, and new divergent isolates might exist. We believe that HTS technologies can be a very good alternative for routine indexing. However, it is still more expensive than two PCRs.

Lane 367: Again, why not testing the sensitivity of the primers instead of predicting with no data to support this? Please see comment above (Lane 350-352).

Lane 371: Another contradiction! here you say inosine can overcome high genetic diversity but on lane 362 you say that inosine do not amplify all divergent sequences. Thank you for your comment. Please see the comment above. (Lanes 360-365). It is not a contradiction, it is a nuanced discussion and hypothesis, with reference to literature.

Conclusions

Lane 386: There is no data to support mutation rates. Pease delete this. Done.

Lane 389: most tests were done in silico. Why didn’t you analyse (in the lab) the samples used in this study with all published primers described on suppl. file S2? It could give a much better comparison of which primer set is the best.

Lane 393: there are many contradictions throughout the manuscript about the utility of the inosine. This needs to be corrected. I think the new primer BanMMV CP9 does not improve the diagnostics of BanMMV and you’re trying to justifying it. If inosine can overcome the high genetic diversity of the virus why did it failed to detect 3 isolates that were positive with primer BanMMV CP2.

Please see the comment above about the utility of the inosine. This base was also used in PDO primers design. This is not the first time  an interest is shown for Inosine in order to design primers for BanMMV. More clarification have been added in both Discussion and Conclusion parts.

Lane 398: this will increase the costs/test. Do you think is the best way to perform diagnostics for BanMMV? Lane 406: I agree you need to design a better primer that detects all diversity. the BanMMV CP9 does not solve your problems and in my view does not improve diagnostics.

We believe that the new primer can be used to improve diagnostic regarding BanMMV (as a complementary analysis). It can be used to limit the false negative rate. We agree that it has some limitations, like all previously designed and published BanMMV primers (a quick reminder that 2 primer pairs  failed outright to bind to new genomes, without going further in the molecular test). This confirmed the high genetic diversity of the virus. Another issue is that the diversity of the viral species seems relatively constant over the genome of the virus. Thus, it is highly challenging (to not say impossible) to have a single primer pair that could recognize all the isolates of the virus in a single PCR. In the same context, we believe the best technique that could be used actually to detect reliably the virus is HTS method even though their application in routine diagnostics will requires thorough evaluation.

Reviewer 3 Report

Dear Authors,

            I had a great opportunity to review research manuscript “Phylogenomics and HTS reveal further diversity and highly variable isolates among viruses in banana” (manuscript id: pathogens-995535) which is considered for publication in Pathogens journal. I analyzed whole manuscript and it showed some interesting insights in topic of some types of banana viruses. In my opinion article need major revision reason for that decision is present in form of specific comments.

  1. Title section

First of all I strongly suggest to add in manuscript title names of analyzed viruses. Currently title is too laconic in my humble opinion.

  1. Abstract section

            All abbreviation used in this section must be explained. If the will be explained here in rest of manuscript authors can freely use abbreviations. Authors must add subfamily and genus for BanMMV according systematic rapport from 2019. Systematics used currently by authors in whole manuscript is inaccurate.

  1. Introduction section

Authors must formulated clear aim of the study with use of statement “Aim of the study was…”.

  1. Materials and methods

            This section is extremely problematic because this section is lack of major amount of needed information. In plant material subsection authors must describe exact conditions of cultivation in greenhouse. According journal publication rules in all used equipment authors must add information about producer and country used software and version In RNA extraction subsection authors must describe data about producer and country Agilent 2100 Bioanalyzer. Immunosorbent electron microscopy (ISEM) must be rewrite completely to show exact procedure because only one cited work is review work completely not associated with this method. All information about antibody dilutions must be added and whole procedure. Name of producers of all antibodies (according journal rules) type of antibody. Why authors used mixture of all antibodies at once? This is extremely strange. How authors eliminated cross-combining of antibodies in to not reacting mass in mixture.  Because effect of cross-reactivity is induced. Is well known that in ISEM in Plant virus research should be performed separately for each type of antibodies. So it must be do it on 3 different grids each grid treated by another type of antibody. Authors did not performed and that error is also seen in results section.  Why subsection 2.5 has not any citations? Is this method of library construction is completely new ? If yes it must be described more. All bioinformatics programs used for phylogeny must have source http but http domains are also literature positions for example in  line 145: “The Geneious software v10.2.6 (https://www.geneious.com)” should be The Geneious software v10.2.6 (Producer and country)[25] where [25] is in reference as (https://www.geneious.com) with date of access.

  1. Results section

ISEM results is problematic. If this technique was performed according rules then particle BanMMV should have black halo around particles because of reaction of BanMMV appropriate antibodies targeted BanMMV. But authors used mixture of antibodies and eliminated detection of specific particles because antibodies cross-reacted. And therefore on Figure 1 particle has no halo and results are not specific. Figure 1 “Filamentous” should not have be written by capital letter. I also strongly recommend to add Photos from separate use of different type of antibodies from higher magnifications. MT872724 and MT872725 sequences are cannot be checked in NCBI Database so information that they are publicated is not true. I strongly suggest to add information of phylogenetic trees information about geographic origin of sequences. Figure 3 must have scaling mark like Figure 2. If authors compare only RdRp fragments from whole genome in tree then after access number of sequences in bracket authors should show range of nucleotides which code this sequence. Authors write in part of results about similarity of amino acids but no trees or other data show in manuscript this analyses.

Sincerely,

Author Response

Reviewer3 :

I had a great opportunity to review research manuscript “Phylogenomics and HTS reveal further diversity and highly variable isolates among viruses in banana” (manuscript id: pathogens-995535) which is considered for publication in Pathogens journal. I analyzed whole manuscript and it showed some interesting insights in topic of some types of banana viruses. In my opinion article need major revision reason for that decision is present in form of specific comments.

Thank you very much for your feedback and constructive remarks. We have taken into consideration all your comments. Please find enclosed the answers to these remarks, point by point. All modifications were highlighted in yellow in the whole manuscript.

PLEASE SEE ATTACHMENT. 

Sincerly, 

Round 2

Reviewer 2 Report

I would like to thank Authors for addressing all reviewers comments. All edits have improved the manuscript. I have no more questions and therefore I am recommending the manuscript for publication. Congratulations.

Reviewer 3 Report

Dear Authors,

The manuscript is improved.